# Implementation barriers and facilitators for referral from the hospital to community-based lifestyle interventions from the perspective of lifestyle professionals: A qualitative study

Leonie M. te Loo [1,2,3], Jasmijn F. M. Holla [3,4], Joyce Vrijsen [5], Anouk Driessen [1,2], Marlinde L. van Dijk [1,2,6], Lilian Linders [3], Inge van den Akker-Scheek [5], Adrie Bouma [7], Leah Schans [8], Linda Schouten [9], Patrick Rijnbeek [10¤], Rienk Dekker [11], Martine de Bruijne [1,2,6], Hidde P. van der Ploeg [1,2], Willem van Mechelen [1,2], Judith G. M. Jelsma [1,2,6]*, on behalf of the LOFIT consortium [¶]

1 Amsterdam UMC, Department of Public and Occupational Health, Vrije Universiteit Amsterdam, Amsterdam, The Netherlands, 2 Amsterdam Public Health Research Institute, Health Behaviors & Chronic Diseases, Amsterdam, The Netherlands, 3 Centre of Expertise Prevention in Care and Wellbeing, Faculty of Health, Sports and Social Work, Inholland University of Applied Sciences, Haarlem, The Netherlands, 4 Reade, Amsterdam Rehabilitation Research Centre, Amsterdam, The Netherlands, 5 University Medical Center Groningen, Department of Orthopedics, University of Groningen, Groningen, The Netherlands, 6 Amsterdam Public Health Research Institute, Quality of Care, Amsterdam, The Netherlands, 7 University Medical Centre Groningen, Department Policy Development in Healthcare Relations, University of Groningen, Groningen, The Netherlands, 8 Huis voor de Sport Groningen, Groningen, The Netherlands, 9 Team Sportservice, Halfweg, The Netherlands, 10 NL Actief, Ede, The Netherlands, 11 University Medical Centre Groningen, Department of Rehabilitation, University of Groningen, Groningen, the Netherlands

¤ Current address: JOGG, Den Haag, The Netherlands
¶ Membership of the LOFIT Consortium is provided in the Acknowledgments.
* j.jelsma@amsterdamumc.nl

## Abstract

### Purpose

A lifestyle front office (LFO) in the hospital is a not yet existing, novel concept that can refer patients under treatment in the hospital to community-based lifestyle interventions (CBLI). The aim of this study was to identify implementation barriers and facilitators regarding the implementation of an LFO in the hospital from the perspective of CBLI-professionals and to develop evidence-based implementation strategies to reduce these identified barriers.

### Methods

We conducted semi-structured interviews until data saturation, with 23 lifestyle professionals working in the community. A semi-structured topic guide was used, inquiring about (1) community-based lifestyle interventions; (2) their views about referral from the LFO; and (3) their preferences, needs and recommendations for collaboration with the LFO in the hospital. The online interviews lasted on average 46 minutes, were audio-recorded and transcribed verbatim. A thematic content analysis was conducted. Found barriers and

**Data Availability Statement:** All relevant data are contained within the article and its Supporting

information files. Within Amsterdam UMC, it is mandatory to delete audio recordings after transcription and verification. The Dutch transcripts of the interviews are available upon reasonable request due to ethical and legal restrictions under the General Data Protection Regulation (EU). The study participants did not consent to making the full transcripts of interviews publicly available. Excerpts of the raw data (quotes) are available within the text of this paper. Requests for anonymized excerpts from the full transcripts can be sent to the general email address: lofit@amsterdamumc.nl.

**Funding:** This research is part of the LOFIT-project which is funded by The Netherlands Organization for Health Research and Development (ZonMw), grant agreement no. 555003208. Also, the national knowledge and innovation network "Centre Of Expertise for Prevention in Care and Welfare" of the Inholland University of Applied Sciences (Hogeschool Inholland) funded this research. The funders of this study (ZonMw & Inholland university of Applied Sciences) had no role in study design, data collection and analysis, decision to publish, or preparation of the manuscript.

**Competing interests:** The authors have declared that no competing interests exist.

facilitators regarding the LFO where mapped using the consolidated framework for implementation research (CFIR) whereafter evidence based strategies were developed using the CFIR-Expert Recommendations for Implementing Change Strategy Matching Tool V.1.0 (CFIR-ERIC).

## Results

Barriers and facilitators were divided into two main themes: 1) barriers and facilitators related to the referral from LFO to CBLI (i.e. financial, cultural, geographical, quality) and 2) barriers and facilitators related to the collaboration between LFO and CBLI (i.e. referral, communication platform and partnership). Thirty-seven implementation strategies concerning 15 barriers were developed and clustered into six overarching strategies: *identify referral options*, *determine qualifications lifestyle professionals*, *develop support tools*, *build networks*, *facilitate learning collaboratives*, and *optimize workflow*.

## Conclusions

In this study, barriers and facilitators for the development of the LFO were found and matching implementation strategies were developed. Practical improvements, like identifying specific referral options or develop support tools, can be implemented immediately. The implementation of other strategies, like connecting care pathways in basic services, primary, secondary and tertiary care, will take more time and effort to come to full potential. Future research should evaluate all implemented strategies.

## Introduction

Healthy lifestyle is key in the prevention and treatment of non-communicable, lifestyle-related diseases. Unhealthy behaviors such as smoking, sedentary and physically inactive lifestyle, unhealthy diet, consumption of alcohol and/or excessive stress are important contributors to the development of non-communicable diseases (NCDs) [1]. When people have already developed an NCD, healthy lifestyle changes can in addition to medical treatment improve health, life-expectancy and quality of life [2, 3], resulting in both personal and societal benefit [4]. Therefore, patients should be encouraged to change their lifestyle when there is opportunity to do so.

A hospital visit provides a window of opportunity to target patients with an unhealthy lifestyle and to convey the importance of a healthy lifestyle, as patients are more motivated for a lifestyle change when experiencing a health event [5–7]. Also, most patients consider the hospital a good location to target modifiable behavioral health risk factors [8]. However, changing one's lifestyle behavior is a difficult task, especially for patients with a lifestyle-related NCD who often have long history of unhealthy lifestyle [9]. Health behavior is the result of several interacting determinants, such as information factors and preceding factors that influence a person's motivation, awareness & action skills [10]. Guidance by a trained lifestyle professional may help patients to stay focused and motivated, to make healthy choices, and to execute a plan for sustained behavioral lifestyle modification [11, 12]. A referral from the hospital to lifestyle coaching in the direct vicinity of the residential environment of the patient provides an accessible and feasible intervention option and is therefore expected to be more effective and less costly than lifestyle consults in the hospital [13]. Furthermore, lifestyle change options are

rarely discussed during hospital consultations, because health care professionals in the hospital often lack the time, knowledge, and required level of motivational skills to encourage their patients to change their lifestyle or seek such interventions in the community [14].

A dedicated lifestyle front office (LFO) in the hospital is a novel concept that may be an appropriate solution for referring patients with NCDs, and those at risk of developing them, to community-based lifestyle interventions (CBLI). The idea of this novel concept is that the treating healthcare professional in the hospital will refer the patient to an appointment with the hospital-based LFO. The healthcare professional in the LFO (i.e. a so-called *lifestyle broker*) will have time, motivational interviewing skills and knowledge of lifestyle referral options in the community. The appointment can be either in the hospital or held via video-appointment, at the convenience of the patient. During the appointment, lifestyle broker and patient will discuss the lifestyle change goal for which the patient is motivated and debate which referral option is most appropriate, given the opportunities, capabilities [15], preferences and needs of the patient and the availability of referral options in the neighborhood of the patient. Referrals from the LFO can be made to CBLI that are part of basic services (e.g. sports club, district social worker, care sport connector, health & fitness centre, lifestyle coach) and primary care (e.g. dietician, psychologist, physical therapist, occupational therapist, lifestyle coach). After the referral, the CBLI-professional regularly inform the lifestyle broker of the patients' progress. Establishing the exact procedure of the referral process is part of this research, in order to align as much as possible with daily practice and preferences of the CBLI-professionals.

For successful implementation and adoption of this yet not existing LFO it is important to involve all stakeholders. In this study, our aim was to identify barriers and facilitators from the perspective of the CBLI-professionals and to develop implementation strategies to overcome these barriers.

## Methods

### Study design

This study was part of the LOFIT-project, in which a lifestyle front office is developed, implemented and evaluated in Dutch hospitals before nation-wide scale up. In this study we performed semi-structured interviews with various CBLI-professionals in the catchment areas of two university medical centers in the Netherlands A qualitative study-design is appropriate for exploring views, needs and preferences of respondents, due to expected time constraints of the professionals, we chose interviews over e.g. focus groups. The COREQ checklist for reporting qualitative research was used as guideline for reporting methods and results [16]. The study was exempted from review by the medical ethical committee of VU University Medical Centre (VUmc) Amsterdam The research does not fall within the scope of the Medical Research Involving Human Subjects Act (WMO), because the research does not have a medical-scientific question.

### Setting

This study was conducted in the Netherlands. Fig 1 provides an overview of the Dutch healthcare system. In the Netherlands, high-quality health care services are equally accessible for everyone and financed through solidarity by taxation of income and mandatory medical insurance [17]. The implementation of Dutch health care is fully determined by five health acts and several health care laws. The Dutch government decides on which health care is covered by the mandatory basic insurance and what should be organized by different parties involved in this system, such as municipalities, health care insurance companies, health care professionals and civilians [18]. The practical organization of health care in The Netherlands is divided in sectors

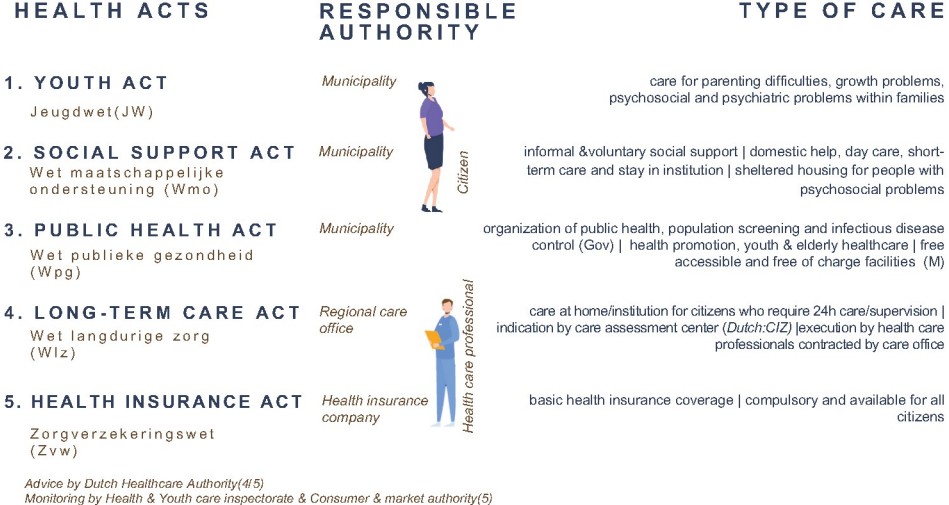

**DUTCH HEALTHCARE SYSTEM**

the Dutch government creates conditions for optimal
public health through 5 health acts

| HEALTH ACTS | RESPONSIBLE AUTHORITY | TYPE OF CARE |
|---|---|---|
| **1. YOUTH ACT** Jeugdwet(JW) | *Municipality* | care for parenting difficulties, growth problems, psychosocial and psychiatric problems within families |
| **2. SOCIAL SUPPORT ACT** Wet maatschappelijke ondersteuning (Wmo) | *Municipality* | informal &voluntary social support | domestic help, day care, short-term care and stay in institution | sheltered housing for people with psychosocial problems |
| **3. PUBLIC HEALTH ACT** Wet publieke gezondheid (Wpg) | *Municipality* | organization of public health, population screening and infectious disease control (Gov) | health promotion, youth & elderly healthcare | free accessible and free of charge facilities (M) |
| **4. LONG-TERM CARE ACT** Wet langdurige zorg (Wlz) | *Regional care office* | care at home/institution for citizens who require 24h care/supervision | indication by care assessment center (*Dutch:CIZ*) |execution by health care professionals contracted by care office |
| **5. HEALTH INSURANCE ACT** Zorgverzekeringswet (Zvw) | *Health insurance company* | basic health insurance coverage | compulsory and available for all citizens |

*Advice by Dutch Healthcare Authority(4l 5)*
*Monitoring by Health & Youth care inspectorate & Consumer & market authority(5)*

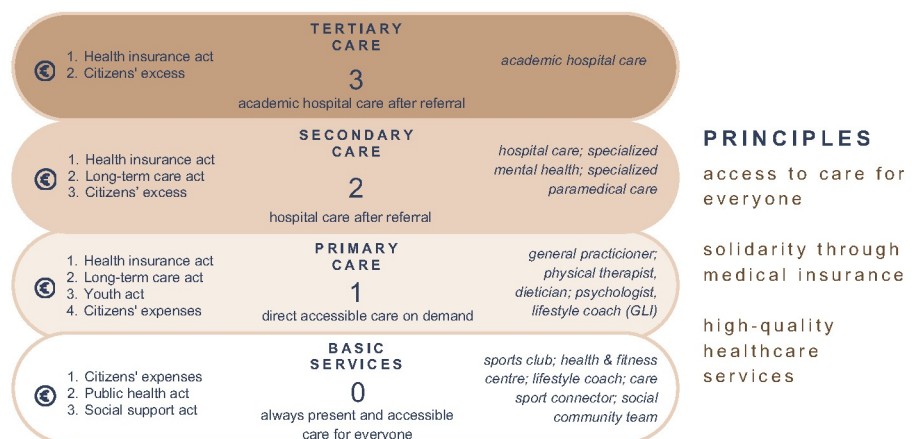

**ORGANIZATION OF CARE**

Dutch health care is organized in four lines to optimize care
consumption and to minimize health costs

| | | | PRINCIPLES |
|---|---|---|---|
| 1. Health insurance act 2. Citizens' excess | **TERTIARY CARE** 3 academic hospital care after referral | *academic hospital care* | |
| 1. Health insurance act 2. Long-term care act 3. Citizens' excess | **SECONDARY CARE** 2 hospital care after referral | *hospital care; specialized mental health; specialized paramedical care* | access to care for everyone |
| 1. Health insurance act 2. Long-term care act 3. Youth act 4. Citizens' expenses | **PRIMARY CARE** 1 direct accessible care on demand | *general practicioner; physical therapist, dietician; psychologist, lifestyle coach (GLI)* | solidarity through medical insurance |
| 1. Citizens' expenses 2. Public health act 3. Social support act | **BASIC SERVICES** 0 always present and accessible care for everyone | *sports club; health & fitness centre; lifestyle coach; care sport connector; social community team* | high-quality healthcare services |

**Fig 1. Dutch health care system.**

in which the government attempts to create optimal health conditions and in which citizens are responsible for their own health if possible. If needed, professional help can be referred to and when required, specialized care is available after referral.

## Participants

The convenience sample for this study consisted of health care professionals providing lifestyle interventions in the community (e.g. lifestyle coaches*, dieticians, stop-smoking coaches, care

## Box 1. Explanation of professions.

### The care sport connector in the Netherlands

A national legislation to stimulate sport participation was installed by the Dutch government in 2008 ['*brede regeling combinatiefuncties*']. Municipalities can apply for co-funding from the government to install a care sport connector in social- and sport organizations, which role can be adapted to local needs but which ultimate goal is to enhance sports participation of all citizens [19]. The care sport connector is also referred to as 'community sports coach' [20].

### Health & fitness centres

Health & fitness centres ("preventiecentra") are specialized sport clubs, in which patients with chronic diseases can work on improving their lifestyle together with specifically trained coaches. Only when joined and verified by the organization NL-Actief, a sport club receives the hallmark "preventiecentrum".

### Lifestyle coach

The title 'lifestyle coach' is not legally protected, although there exists a professional association (Association for Lifestyle Coaches in the Netherlands (ALCN, Dutch abbreviation BLCN) which manages and controls quality certification for lifestyle coaches. Only a ALCN-certified lifestyle coach may execute one of the seven *Combined Lifestyle interventions* (CLI). These two-year programs are covered by basic health care insurance. Lifestyle coaches may work for organizations which stimulate health improvement by means of lifestyle change; they can also work for organizations which use lifestyle changes as a means to improve other issues (e.g. loneliness).

sport connectors*, physical therapists and professionals working in a health & fitness centre*) to include a broad variety of perspectives. See Box 1 for an elaboration on some of the professions (*). The goal was to recruit participants until thematic data saturation.

On the internet we searched for the above-mentioned health care professionals and relevant organizations, in the wider area around the two university medical centers in the Netherlands. The recruitment period lasted from July 6th until December 6th, 2021. Potential respondents were approached and informed about the study by email and were asked to participate. If they responded positively, an online video call of 45 minutes was scheduled via Microsoft Teams at their convenience. In case we received no response within a week, one reminder was sent. Furthermore, snowballing was applied by asking participants for other potentially interested health care professionals in their network. All participants gave written informed consent prior to the interview.

## Data collection

To identify barriers and facilitators regarding implementation of and collaboration with the LFO we inquired about three topics: (1) information about the CBLI in which the professional participates; (2) their views about the concept of an LFO in the hospital; and (3) their preferences, needs and recommendations for collaboration with such an LFO. A semi-structured topic guide was used for the interviews (S1 Appendix). The guide included five sections with open questions to start each topic and further prompts to gain more in-depth insights. Depending on the job title of the respondent some sections were skipped. For instance, section 3 and 4 contained specific questions for the care sport connector which were not applicable to the lifestyle coach. The interview was started by the interviewer introducing herself and her role in the study. Thereafter, participants were asked to introduce themselves with regard to their profession. Because the LFO in the hospital is part of a completely new care pathway, a short introduction to the background and purpose of the LFO was given by the interviewer before discussing this topic.

Interviews were conducted in Dutch by LL (MSc), AD (MSc), JJ (PhD) and JV (PhD), between July and December 2021. All interviewers had previous experience in conducting qualitative interviews and were employed as junior researcher (LL, AD) or senior researcher (JV, JJ). All interviews were held online due to nationwide COVID-19 restrictions. No other persons were present during the interview. Interviews were audio-recorded with the explicit consent of the participant, using a voice recorder. When a topic was finished, the interviewer summarized the conversation and the respondent was asked if this summary was accurate and if they wanted to add information. After each interview, a summary was written immediately by the interviewer, to capture the most important findings and to use the summary during the analyzing process to be able to verify codes or themes. Interviews were conducted until thematic data saturation [21], which was established after 23 interviews when investigators found that no new themes emerged from the interviews.

## Data analysis

Recordings of interviews were transcribed verbatim, using automated transcription software (Amberscript, Amsterdam, The Netherlands) with manual correction. Thematic content analysis was conducted in software package MAXQDA 2020. Thematic content analysis is a useful approach in identifying key elements of participants' accounts, which matches the aim of our study [22]. We used the five steps of the thematic content analysis method to identify, analyze and report overarching themes: 1) *compiling* (i.e. collecting, familiarization and transcription of data); 2) *disassembling* (i.e. initial open coding of the data conducted by AD and LL); 3) *reassembling* (i.e. creating a codebook with definitions of (sub)themes in which codes were clustered based on consensus during a coding meeting with AD, LL, JJ, JH (S2 Appendix), where after all interviews were (re)coded with this codebook by LL, AD, JV) and summaries of the interviews were re-read to verify that all relevant data was included; 4) *interpreting* (i.e. re-reading of all codes within themes and creating analytical conclusions within and across themes by LL, JJ, JH); 5) *concluding* (i.e. creating answers to the research question(s)) [22, 23]. Data were analyzed in Dutch for all these steps. Dutch is the native language of all involved researchers.

To be able to overcome identified barriers for the referral from the LFO in the hospital to CBLI, we used the consolidated framework for implementation research (CFIR) [24] in combination with the CFIR-ERIC (Expert Recommendations for Implementing Change)-matching tool [25]. The CFIR specifies 37 constructs within five general domains, that are believed to influence implementation. The 73 ERIC implementation strategies are based on consensus

of an expert panel [26]. First, three authors (LL, JJ, MD) separately labeled barriers with one or more CFIR constructs. Thereafter the choices were discussed to ensure appropriate labeling and to reach consensus. After consensus, the CFIR constructs were matched to ERIC-strategies using the CFIR-ERIC-matching tool which ranks the best associated strategies according to expert opinion. We used the top 4 strategies produced by the tool for every set of constructs. Finally, these strategies were discussed and practical translations of the strategies that were close to practice because of the involvement of the research team in the development and implementation of the LFO were formulated (LL, JJ, MD, JH).

## Results

### Interviews

**Participants.** In total 65 CBLI-professionals were approached for participation in the study (*lifestyle coaches (n = 41), dieticians (n = 4), care sport connectors (n = 12), physical therapists (n = 2), account manager quit-smoking intervention (n = 1) and managers of health & fitness centres (n = 5)*). Forty-one professionals did not respond to the first email and the reminder. One participant cancelled the meeting due to a busy work schedule and did not respond to email after that. In total 23 professionals agreed to an interview. The participants were on average 47 years old (median 49; range 25–65) and 65% (n = 15) of the participants was women. Sixty-one percent (n = 14) had less than 5 years of work experience in their current position; 9% (n = 2) had between 5–10 years of experience, 30% (n = 7) had more than 10 years of working experience in the current position. In Table 1 an overview of the characteristics of the participants is provided. The interviews lasted on average 46 min (range: 34–72 min).

**Themes.** We pragmatically arranged barriers and facilitators according to successive events in the LFO, resulting in two main themes: (1) barriers and facilitators related to the referral options from the LFO; in this theme, the results could be further divided into the subthemes a) accessibility of CBLI and b) quality of CBLI; and (2) barriers and facilitators related to the collaboration between the LFO and CBLI, with the subthemes a) the referral content and process; b) communication platform and c) partnership between the LFO and the CBLI. In the next paragraphs results are presented according to these themes and subthemes. A summary of all barriers and facilitators per theme and subtheme is tabled in S3 Appendix.

**Table 1. Characteristics of participants.**

| Total | n = 23 |
|---|---|
| Age in years, median (range) | 49 (25–65) |
| Gender, women, n (%) | 15 (65%) |
| Lifestyle professionals job title * | **n** |
| • Lifestyle coach | 16 |
| • Dietician | 1 |
| • Care sport connector | 3 |
| • Coordinator care sport connectors | 3 |
| • Manager health & fitness centre | 4 |
| • Account manager quit smoking intervention | 1 |
| Relevant working experience in years, median (range) | 4 (1–30) |

* total adds up to more than 23 because some participants have more than one job title

**1. Barriers and facilitators regarding referral options from the LFO.** *1a. Accessibility of CBLI.* Barriers and facilitators were found for financial, cultural and geographical accessibility of the CBLI.

**Financial accessibility.** Financial accessibility of lifestyle interventions is a barrier for the LFO when referring patients with limited financial resources. A referral to a gym or a personal lifestyle coach can be expensive, limiting possibilities to refer patients to such interventions.

"*I see a lot of people who cannot afford to exercise [at a gym or club]*"

[#10; Coordinator care sport connectors, care sport connector & lifestyle coach]

Besides that, it was mentioned that some patients are not used or willing to pay for their own health.

"*It is an additional barrier if people have to pay for it [the referral option]. People with low socioeconomic status or position are not used to spending money on their wellbeing*"

[#15; Lifestyle coach]

However, there are options available for patients with lower income, like exercise activities organized by welfare organizations, that patients can attend for free or at a small fee. Another option is one of the seven evidence-based combined lifestyle interventions (GLI, see box 1), which are covered by basic health care insurance in case a patient is eligible according to certain criteria.

"*The combined lifestyle intervention [5] is fully paid for by basic health care insurance without excess. For some patients that is the reason why they opted for the GLI and not for individual counselling*"

[#3; Dietician & lifestyle coach]

Dietary care is also covered by basic health care insurance for the first three hours. Respondent 3, however, feels that this is not sufficient for patients with complex problems.

"*It depends on the financial resources people have. If not, then you are restricted to three hours insured care. That is too little, you often will not manage with that, because most people have to deal with their eating disorder. I can easily spend five to six hours on that topic. That is not always possible, because not everyone has the financial means for that. This prevents you in executing your job sufficiently.*"

[#3; Dietician & lifestyle coach]

In addition, if someone has already used the basic health insurance to reimburse an attempt to stop smoking, a second attempt in the same year will not be covered. This can be a barrier for the LFO when referring a patient who is motivated to quit smoking, but who has already used the yearly insurance coverage.

"*For instance, if we are contacted by someone in October who says: "I participated [in your program] earlier this year. At that time, I was not successful, but I want to try again now". We have to inform the patient that the insurance only covers one attempt per year. Of course, they*"

*may still join [the program], but they have to pay for it themselves. Then they say: "I will wait until next year."*

[#23 Account manager quit smoking intervention]

As a facilitator it was mentioned that municipalities might provide additional local funding opportunities for people with low income. Knowledge of such funding opportunities could facilitate patient referral.

"*I have been fortunate that I can contact a civil servant "work and income" directly: If someone says: I really want support, but I cannot afford it", I redirect that question to the civil servant immediately.*"

[#6; Manager Health & fitness centre]

**Cultural accessibility.** Language can be a barrier for people who do neither speak Dutch nor English. When these patients are referred to Dutch- or English-speaking lifestyle professionals, communication is complicated, which might influence the intervention.

"*You see the same thing [not speaking Dutch] with Turkish or Moroccan people, which is of course difficult. I see women suffer from obesity who do not always speak Dutch, which makes it [the conversation] tough. A son or daughter always has to accompany them.*"

[#3; Dietician & lifestyle coach]

Especially group interventions where communication is key, such as the combined lifestyle intervention, are almost exclusively offered in the Dutch language. This may result in non-optimal referrals for some patients. As explained by a lifestyle coach:

"*it would be very unfortunate that a foreign speaker only can be referred to 1-on-1 coaching, even though a combined lifestyle intervention would have been a better option, but that is not possible [because it is in Dutch].*"

[#4; Lifestyle coach]

Knowledge of all existing CBLI who offer guidance in a foreign language is a facilitator for successful referral of foreign speaking patients.

**Geographical accessibility.** Care sport connectors indicated that it will be hard for an LFO in the hospital to remain geographically up to date regarding all available referral options in the field of physical activity and sports, because locally there is a large and continuous changing variety of options.

*It is almost undoable for you [the LFO] to find all sports and physical activity interventions in the district of the care sport connector.*

[#14; Manager care sport connectors]

However, in the Netherlands health & fitness centres provide exercise programs for people with different health conditions nationwide.

"*Currently, there are about 80–90 sports centers applying for the 'preventiecentrum' hallmark with NL Actief. A big advantage is that we have coverage in the whole of the Netherlands*"

[#8; Manager health & fitness centre]

The care sport connectors indicated to be willing to serve as intermediary partner in the field of sports and physical activity. This will be a facilitator for the LFO to help find a fitting referral for patients with physical activity goals. Some care sport connectors indicated that it is already part of their job description to support clients in their district towards sport participation. However, it was mentioned that the care sports connectors are organized differently everywhere, which may be a barrier for the LFO to find them and work together.

"*. . . in one municipality, the care sport connector is employed by the municipality, in another by a local sports organization, and yet another by a welfare institution*

[#14; Manager care sport connectors]

It was mentioned that care sport connectors may have different tasks, depending on the local implementation of (national) policy and the hosting organization. Some care sport connectors have a connecting role between the different sports organizations, local health care, the municipality and its citizens in order to enhance sports participation or even to create social impact, other care sport connectors' only task is to organize and execute activities with certain groups.

*And it is also what you [the LFO] want from a care sports connector. Is it within his task? Can it be done in the available time? The strength of the national policy is that you can customize [the role of the care sport connector] to local needs. But that is also its weakness, because there is no unity in the role.*"

[#14; Manager care sport connectors]

*1b. Quality of CBLI.* Respondents feel the need for the LFO to build a network of dedicated, trained professionals in the local community.

"*To create a network with lifestyle coaches or other people that you know have a certain expertise. That is useful I think.*"

[#21; Lifestyle coach]

Sports and exercise referral opportunities for patients with specific health conditions might be limited as these patients need more specialized guidance.

"*Also, I think the University Medical Centers will mostly refer people who are overweight or have a more complex medical status. I do not know if we can respond to everybody's [supervision] needs.*"

[#10; Coordinator care sport connectors, care sport connector & lifestyle coach]

Some lifestyle professionals do not have a medical background, which could be a barrier when receiving referrals of patients with medical conditions.

*"Some lifestyle coaches have a completely different background with no medical expertise. I do not know how you can select on this, but if you are referring people with medical conditions, it seems to me something you should consider."*

[#3; Dietician & lifestyle coach]

Respondents who are working in the fitness industry feel that primary and secondary care do not always see them as quality partners, because their main interest is thought to be financial.

*"Primary and secondary care have a certain view regarding the fitness industry. We are commercially oriented and only think about gaining more members. This has changed long ago, but this stigma still exists."*

[[#6; Manager health & fitness centre]

However, health & fitness centres indicate that they are keen to accommodate people with different health conditions and they feel that they are prepared and trained for this.

*"[In our center] we see oncological patients, people who are overweight, COPD patients, name it- we have protocolled interventions for all these groups"*

[#8; Manager health & fitness centre]

**2. Barriers and facilitators for collaboration between the LFO and the CBLI.**    *2a. Referral content and process.* The accountable person for the referral of the patient from the LFO to the CBLI is a topic on which respondents have differing opinions. Some mentioned that to encourage patient self-management, the patient should initiate the appointment with the lifestyle intervention themselves.

*"I think you should give the people autonomy; we all do that in our lifestyle programs. We make sure, that the people who are in our trajectory, maintain control over their lives, their own changes, etc. We do not say: "you will have to do this or that", that does not work, not at all. Eventually they have to do it by themselves, right? So, if I could decide, I think you should let the participant choose [who makes the appointment with the CBLI]."*

[#8; Manager health & fitness centre]

Others endorse the importance of stimulating self-management, but argue from their experience that patients do not make this appointment and need some form of reminder or assistance. They suggest that the LFO should contact the community-based lifestyle professional, who in turn should actively contact the patient.

No, that is something we do [making an appointment]. Otherwise, nothing will happen. Yes, that is really our experience, also because we are talking about a group that does not address these issues by themselves easily.

[#15; Lifestyle coach]

Respondents state that they are not medical experts and are not primarily educated to deal with complex medical problems. If there are specific medical considerations regarding the patient, they should be made explicit with the referral.

*"Yes, all medical information that could be relevant for a lifestyle coach. And the possibility to contact the referrer in case of doubt, because we are not educated to make certain medical decisions."* [#1; Lifestyle coach]

The allocation of tasks in the collaboration must be coordinated and, according to the participants, is a shared responsibility between the LFO and a CBLI.

*"Collaboration is a shared responsibility, but I think one that certainly could be initiated by the [lifestyle] providers themselves."*

[#9; Lifestyle coach]

*2b Communication platform.* In primary and secondary care it is common to communicate confidential patient information via secure e-mail with other professionals. However, according to respondents, this does not work between primary and secondary care, which is a barrier for collaboration between the LFO and the CBLI.

*"Up until now my experience is that hospital physicians do not have a "ZorgMail' [secure mail] account. For us it is then not possible to give feedback on a single patient."*

[#23; Account manager quit smoking intervention]

For effective collaboration, respondents want to be able to share confidential information in a secure way with the LFO and vice versa.

*"The medical confidentiality. What information do you share and what not? Although, probably through a digital secured system, you should be able to share this with other health care professionals."*

[#21; Lifestyle coach]

*"[. . .], although you will need a well secured system [for communication]. It is not directly available. You [the hospital] will have to invest in that development."*

[#22; Lifestyle coach]

There are different online communication platforms available for secure digital communication and many participants are already familiar with a number of them. However, the use of different platforms is sometimes complicated and tedious, but most are glad to have a solution for secure communication with other care providers.

*"Am I glad to work with four different systems? No, but if it cannot be different, then we will just have to manage."*

[#8; Manager health & fitness centre]

Most platforms require a paid subscription in order to use it for secure communication. Respondents who work for a healthcare institution can often use the IT-system of the institution to communicate securely, while self-employed professionals mostly do not have a

connection or subscription to such secure IT-platforms. The relative costs are too high to make use of it. This will be a barrier for the LFO if paid IT systems are used in the communication between LFO and CBLI.

> "*I guide two groups. If I add the financial costs for license fees, for the program and my accreditation, and then also for a secure digital platform, then there is no profit from guiding these groups.*"
>
> [#11; Lifestyle coach]

*2c Partnership between the LFO and the CBLI.* Respondents are eager to work together with the LFO, although most respondents have little to no experience in collaborating with the hospital. The ones that do have such experience feel that primary and secondary care work segregated. Consequently, they feel that this segregation is a barrier for collaboration between the LFO and the CBLI.

> "*But I think that primary and secondary care are still two separate care pathways, [. . .]. Although I believe the hospital should keep tabs on it [the collaboration].*"
>
> [#22; Lifestyle coach]

For effective and continued collaboration, lifestyle professionals find it important that they personally know the lifestyle broker in the LFO who refers patients to them. The possibility of personal, direct communication is necessary to understand each other, especially at the start of the collaboration. Also, they think it will help the patient trust the CBLI more.

> "*I noticed it is all about trust. Then they [the patients] trust the [care delivered in] hospital and not [care delivered in] primary care. Such a relationship of trust and effective handoff is very important. When the participant notices: 'I am now with this [primary care] lifestyle coach, who is known by the lifestyle broker, than it all feels secure.*"
>
> [#22; Lifestyle coach]

The feeling of being part of a network and collaborating on the same patient case is also very important to lifestyle professionals, as is recognition of their expertise.

> "*Yes, make them [the health & fitness centre employees] feel that they are a part of the care the patient receives in the hospital. We seek recognition of the part we play in the care pathway. Not 'just go there', but, we are part of the whole process.*"
>
> [#6; Health & fitness centre manager]

## Implementation strategies

From the CFIR-ERIC-matching tool we gained 37 implementation strategies to overcome 15 identified barriers. All barriers, CFIR constructs, ERIC strategies and operationalized strategies for the implementation of the LFO with regard to the CBLI are presented in S4 Appendix. We clustered the strategies into six overarching strategies (Table 2): *identify referral options, determine qualifications lifestyle professionals, develop support tools, build networks, facilitate learning collaboratives, and optimize workflow.*

**Table 2. Strategies per barrier.**

| Barriers perceived by CBLI | Strategies | identify referral options | determine qualifications lifestyle professionals | develop support tools | build networks | facilitate learning collaboratives | optimize workflow |
|---|---|---|---|---|---|---|---|
| **Financial accessibility** | | | | | | | |
| Lifestyle interventions can be too expensive for patients with low income | | xx | | | | | |
| Health care insurance coverage is not enough for needed care | | x | | | | x | |
| Applying for funding is difficult for some patients | | x | | x | x | | x |
| **Cultural accessibility** | | | | | | | |
| Few referral options for people who do not speak English or Dutch/Combined lifestyle intervention offered almost exclusively in Dutch language | | xx | | | | | |
| **Geographical accessibility** | | | | | | | |
| Job descriptions of care sport connectors vary which makes it harder to use them as intermediary (not all are suitable) | | | x | x | | | x |
| Lifestyle coaches specialized in complex medical patients may not always be present in the field of exercise | | | xx | x | | | |
| It is hard for the LFO to gain overview of the large and continuous changing variety in referral options | | | | x | x | x | |
| Care sport connectors are employed by different types of organizations which hampers the findability of the care sport connector | | | | | | x | |
| **Quality of referral options** | | | | | | | |
| Some lifestyle coaches do not have medical background | | x | | x | | x | x |
| Fitness industry feels that primary and secondary care do not see them as quality partners | | | x | | | x | |
| **Referral content & process** | | | | | | | |
| Lifestyle professionals believe that patients do not make an appointment with the CBLI on their own | | | | | | | x |
| **Communication platform** | | | | | | | |
| The use of different systems is confusing for the CBLI | | | | x | | | xx |
| Secure email does not work between hospital and primary care/Without secure communication it is not possible to give feedback of the patient's progress | | | | | | x | x |

*(Continued)*

**Table 2.** (Continued)

| Barriers perceived by CBLI | Strategies | identify referral options | determine qualifications lifestyle professionals | develop support tools | build networks | facilitate learning collaboratives | optimize workflow |
|---|---|---|---|---|---|---|---|
| Costs of paid communication systems are too high for the CBLI | | | | | | | x |
| **Partnership** | | | | | | | |
| Primary and secondary care work segregated | | | | | x | xxx | |

Each "x" denotes a specific singular strategy. xx/xxx means two/three different operationalized strategies were formulated for this barrier in this cluster.

**Identify referral options.** For the LFO to be able to refer the patient to the most appropriate referral option, the following referral options are to be identified explicitly by the lifestyle broker before implementation: low- and no cost lifestyle interventions (for patients with low income), lifestyle interventions in different languages (for patients who do not speak Dutch or English) and the location of CBLI with medical specializations (for patients with complex medical health status) Also, the lifestyle broker should be able to locate benefits and arrangements that exist for residents with low income and also know to which person or organization they can refer the patient for assistance in applying for these benefits and arrangements. During implementation, language adaptation needs of patients in the LFO should be identified for further improvement of the referral options of the LFO

**Determine qualifications lifestyle professionals.** Before implementation, the lifestyle broker profits from gaining more insight into the qualifications of the referral options to refer the complex medical patients to. Also, it is important to assess the conditions on when medical knowledge is needed and if and how this knowledge can be acquired by the CBLI. Because the field of exercise and sports may be too large for the lifestyle broker to keep up with, we need specifically, more insight in the different job profiles of the care sport connectors in order to decide which profile can function as intermediary for the LFO.

**Develop support tools.** Before implementation, it will be helpful if the lifestyle broker develops some support tools. Firstly, a (digital) referral tool which lists referral options will help the lifestyle broker to find qualified CBLI-professionals in the patient's residential area Additionally, an overview of care sport connector-profiles that states which role each profile has in relation to the LFO. For the patient with low income, informational material on how to apply for funding should be developed. Also, in case there is no lifestyle professional with medical background available in the neighborhood of the patient, a template for medical and safety information handover should be made for the referral from the LFO and finally, to make the use of a digital communication platform as easy as possible for the CBLI, a manual on the use of the network communication platform should be developed.

**Build networks.** During implementation, the lifestyle broker should actively work on building a network of partners to keep up with possible referral options and to improve collaborating within and between care pathways. With the CBLI-professionals- this can be initiated by making personal contact during handover and also after some weeks, to ensure patient transfer and adequate referral, but also to be able to document the experiences of the lifestyle broker with the CBLI. The lifestyle broker has contact with health professionals within the hospital as well as the CBLI-professionals in primary care and basic services, which is a unique position to promote collaboration between these different care pathways that are not used to

work together. Additionally, a network of professionals that can assist applying for funding for patients with low income should be built before and during implementation.

**Facilitate learning collaboratives.** During implementation the lifestyle broker will organize and participate in learning collaboratives. This includes intervision meetings between lifestyle brokers of other LFO's, to share experiences regarding referral options, especially referral options in different languages and with low or no costs. Also, the lifestyle broker will organize meetings with CBLI-professionals in which they can learn from each other regarding medical education, findability, digital communication and collaboration between the care pathways. Additionally, the lifestyle broker plans working visits to the CBLI in order to further promote collaboration.

**Optimize workflow.** The devised workflow within the LFO can be optimized by executing the following strategies. In the large field of sports and exercise, it will be helpful to use an intermediary with local knowledge of referral options, such as the care sport connector, to help deciding on the most appropriate CBLI. Representatives of the care sport connectors should be included in the development before the start of the LFO, to think along with the implementation team on how this should be done and what agreements must be made.

Secondly, a secure network communication platform, ideally without costs for the referral partners, is recommended to be used for communication with the CBLI. Because CBLI-professionals find the use of different systems confusing, regional and national trends must be taken into account in order to decide on the best suited platform. Also, the lifestyle broker should assist with the use of the network platform and promote it during implementation with the CBLI.

Thirdly, to make sure the patient makes an appointment with the CBLI the lifestyle broker and patient discuss patient's responsibility in the transfer to the CBLI and when necessary, the lifestyle broker offers assistance to the patient. The referral from the LFO should when possible be done to professionals with a quality hallmark.

## Discussion

In this study our aim was to find barriers and facilitators for the implementation of an LFO in the hospital according to CBLI. Main barriers where the limited referral options for patients with low income, patients who do not speak English or Dutch or patients who live in rural areas. Also, the quality of the referral options is not always known and poses a barrier for the referral. A major barrier is that currently, there is barely any collaboration between basic services, primary and secondary care. In line with this, secure communication between those pathways is challenging. The positive attitude of the CBLI towards collaboration with the LFO is an important facilitator for the LFO, as is the commitment to build a professional but personal relationship with each other. We used the CFIR-ERIC-tool to select effective, actionable strategies [27], which we operationalized into practical strategies for the LFO. Strategies concerned identifying limited referral options and professionals with a quality hallmark and strategies to streamline the actual referral and the process of collaboration between basic services, primary and secondary care.

Referral from the LFO to CBLI is hindered by patient's low income [28], native language (i.e. not Dutch or English speaking) [29, 30] and living area (i.e. rural areas) [31, 32], which limits actual behavioral change support options. This is extra important because the prevalence of NCDs is highest in these groups [33, 34] and speaking the same language is important for behavioral change [35, 36]. We do know that culturally-tailored interventions in which not only language but the whole intervention is adapted to the target group are known to have more impact on changing lifestyle behavior [37–40].

Patient safety in medically complex patients was mentioned by the CBLI as barrier for referral to primary care or basic services, which is also found in previous interviews with health care professionals and general practitioners [41, 42].

Although to behave more healthy in daily life (i.e. eating healthier, being more physically active or quitting smoking) not much harm is involved and for almost every patient these daily activities can be performed safely [43]. However, sometimes the patient can also feel insecure [44, 45] and a quality hallmark or specific program (e.g. lifestyle intervention specific for patients with Type 2 diabetes) might facilitate overcoming this barrier to become active. In future research we should investigate these opposing perspectives to determine in which situation a CBLI needs knowledge about and/or experience with specific patient groups.

Collaboration and integration between multiple disciplines in various care sectors, should be enhanced [46] as previous integrated care programs have shown positive results on quality of patient care [47]. However, the implementation of integrated care, despite its potential, is complex because of different stakes, processes and contextual factors. The Healthy Alliances (HALL) framework [48] states vital components for successful collaboration (i.e. personal relationship, shared identity with common aims and an attitude in which partners trust each other) that are also supported by this study. In a recent review about competencies that are required for health care workers in primary and secondary care to promote collaboration, it was found that a common concern, mutual knowledge, good communication and a collaborative attitude and respect are important [49], which is in agreement with what was found in our interviews.

Digital health can facilitate and improve health care in various applications in order to make disease management and communication in the health care processes more patient-centered [50, 51]. Furthermore, the timely and accurate transfer of referral content facilitates the quality of patient care and can be operationalized through a simple template [52]. Also, in the Netherlands, the Integral Healthcare Agreement (Dutch: IZA) was deployed in which digitization of care is a main prerequisite [46]. Respondents underlined the need for secure communication as a means to optimize collaboration, but are reluctant regarding potential extra work and costs of another new platform.

## Strengths and limitations

A strength of this study is that we used data-triangulation by collecting input from various lifestyle professionals (n = 23), through which we gained insight on the topics from different perspectives. The number of interviews is high, although it should be noted that the number of interviews within the different types of lifestyle professionals was in some cases low. For instance, we only interviewed one dietician. However, in this research we aspired to know the perspectives of CBLI-professionals that the LFO could refer to, and we feel that overall data saturation was achieved. Furthermore, because of our choice to use a convenience sample, it is possible that we interviewed professionals who are eager to promote their views and that we missed more reluctant professionals who may have a different opinion. However, we invited a broad variety of lifestyle professionals based on an internet search and through professional network maps. Data collection and analysis were performed by a group of researchers with varied expertise which enhances the credibility of the findings. Although the study was conducted in the Netherlands, the results are also valuable for other countries to optimize collaboration and referral for lifestyle care between hospitals, primary care and social care services. The interviews were performed in order to develop the LFO. Because the LFO did not yet exist, the topics were sometimes hypothetical and harder to envision for the lifestyle professionals, as they had no experience working with an LFO. The implementation strategies were

largely developed based on previous literature [24–26] and further operationalized based on expertise from the research team, therefore the actual successfulness of the developed implementation strategies and the LFO organization needs to be further tested [53]. Finally, in this study, the external stakeholders were the topic of research. The perspectives of the internal stakeholders of the LFO (health care professionals within the hospital and the patients under treatment in the hospital) have been assessed in another study in the LOFIT-project and will be published separately.

## Conclusions

In this study facilitators and barriers were found for referral from the LFO to CBLI (i.e. financial, cultural, geographical, quality) and for collaboration with CBLI-professionals (i.e. referral, communication platform, partnership). The developed matching implementation strategies aids the development of the LFO in a practical manner (e.g. a referral tool, patient handover template, referral procedure to optimize workflow), and also in a communicational and educational way (e.g. lifestyle broker intervision meetings, CBLI network events). In future research we will evaluate these strategies to draw conclusions about their effectiveness and to optimize them based on advancing insight and experience. This will further reduce challenges in hospital-community collaboration on lifestyle care will further reduce challenges in hospital-community collaboration on lifestyle care.

## Supporting information

**S1 Appendix. Topic guide for interview stakeholders CBLI.**
(PDF)

**S2 Appendix. Codebook.**
(PDF)

**S3 Appendix. Summary of all barriers and facilitators per theme and subtheme.**
B = barrier; F = facilitator.
(PDF)

**S4 Appendix. Overview barriers and matched strategies.**
(PDF)

## Acknowledgments

We thank all participants for their time and effort. Also, we would like to acknowledge the members of the LOFIT consortium for their contribution to this research: Inge van den Akker-Scheek (Dept. of Orthopedics, UMC Groningen), Marjan Annema (Dept. of Orthopedics, Ommelander Hospital Groningen), André van Beek (Dept. of Endocrinology, UMC Groningen), Jip van den Berg (Dept. of Nephrology, UMC Groningen), Alexander L. Boerboom (Dept. of Orthopedics, UMC Groningen), Adrie Bouma (Dept. of Rehabilitation Medicine, UMC Groningen), Martine de Bruijne (Amsterdam Public Health Research Institute), Jeroen Crasborn (Health Insurance Expertise), Rienk Dekker (Dept. of Rehabilitation Medicine, UMC Groningen), Johanna M. van Dongen (Amsterdam Public Health Research Institute), Anouk Driessen (AmsterdamUMC; Amsterdam Public Health Research Institute), Marlinde L. van Dijk (AmsterdamUMC; Amsterdam Public Health Research Institute;), Karin Eijkelenkamp (Dept. of Endocrinology, UMC Groningen), Nies Goelema (Dept. of Orthopedics, Ommelander Hospital Groningen), Jasmijn F.M. Holla (Inholland University of Applied Sciences, Haarlem; Reade Rehabilitation Research Center, Amsterdam), **Judith G.M. Jelsma**

(lead author;.jelsma@amsterdamumc.nl; AmsterdamUMC; Amsterdam Public Health Research Institute), Johan de Jong (Hanze University of Applied Sciences, Groningen), Anoek de Joode (Dept. of Nephrology, UMC Groningen), Arthur Kievit (Dept. of Orthopedics, Amsterdam UMC), Josine van't Klooster (Dept. of Strategy, UMC Groningen) Hinke Kruizenga (Dept. of Nutrition & Dietetics, Amsterdam UMC), Marike van der Leeden (Dept. of Rehabilitation Medicine, UMC Groningen), Lilian Linders (Inholland University of Applied Sciences, Haarlem), Leonie M. te Loo (Inholland University of Applied Sciences, Haarlem; Amsterdam UMC; Amsterdam Public Health Research Institute), Jenny Marks-Vieveen (Dept. of Anestesiology, Amsterdam UMC; Amsterdam Public Health Research Institute), Willem van Mechelen (Amsterdam UMC; Amsterdam Public Health Research Institute), Douwe Johannes Mulder (Dept. of Internal medicine, Amsterdam UMC), Femmy Muller (Health Insurance organisation *Zilveren Kruis*), Femke van Nassau (Amsterdam UMC; Amsterdam Public Health Research Institute), Joske Nauta (Amsterdam UMC; Amsterdam Public Health Research Institute), Suzanne Oostvogels (Health Insurance organisation *Menzis)*, Hidde P. van der Ploeg (Amsterdam UMC; Amsterdam Public Health Research Institute), Patrick Rijnbeek (NL actief, Ede; current address: JOGG, Den Haag, The Netherlands) Leah Schans (Huis voor de sport Groningen), Linda Schouten (Team Sportservice Noord-Holland, Haarlem), Charlotte Schumacher(Dept. of Orthopedics, UMC Groningen), Rhoda Schuling (Hanze University of Applied Sciences, Groningen), Erik H. Serné (Dept. of Internal medicine, Amsterdam UMC), Maarten R. Soeters (Dept. of Internal medicine, Amsterdam UMC), Evert A. L. M. Verhagen (Amsterdam UMC; Amsterdam Public Health Research Institute), Joyce Vrijsen (Dept. of Orthopedics, UMC Groningen), Johannes Zwerver (Human Movement Sciences, UMC Groningen).

## Author Contributions

**Conceptualization:** Leonie M. te Loo, Jasmijn F. M. Holla, Joyce Vrijsen, Inge van den Akker-Scheek, Hidde P. van der Ploeg, Willem van Mechelen, Judith G. M. Jelsma.

**Data curation:** Leonie M. te Loo, Joyce Vrijsen, Inge van den Akker-Scheek, Judith G. M. Jelsma.

**Formal analysis:** Leonie M. te Loo, Jasmijn F. M. Holla, Joyce Vrijsen, Anouk Driessen, Marlinde L. van Dijk, Judith G. M. Jelsma.

**Funding acquisition:** Jasmijn F. M. Holla, Lilian Linders, Inge van den Akker-Scheek, Adrie Bouma, Leah Schans, Linda Schouten, Patrick Rijnbeek, Rienk Dekker, Martine de Bruijne, Hidde P. van der Ploeg, Willem van Mechelen, Judith G. M. Jelsma.

**Investigation:** Leonie M. te Loo, Joyce Vrijsen, Anouk Driessen, Judith G. M. Jelsma.

**Methodology:** Leonie M. te Loo, Jasmijn F. M. Holla, Joyce Vrijsen, Anouk Driessen, Marlinde L. van Dijk, Inge van den Akker-Scheek, Hidde P. van der Ploeg, Willem van Mechelen, Judith G. M. Jelsma.

**Project administration:** Leonie M. te Loo, Jasmijn F. M. Holla, Joyce Vrijsen, Hidde P. van der Ploeg, Willem van Mechelen, Judith G. M. Jelsma.

**Supervision:** Jasmijn F. M. Holla, Hidde P. van der Ploeg, Willem van Mechelen, Judith G. M. Jelsma.

**Visualization:** Leonie M. te Loo.

**Writing – original draft:** Leonie M. te Loo.

**Writing – review & editing:** Leonie M. te Loo, Jasmijn F. M. Holla, Joyce Vrijsen, Anouk Driessen, Marlinde L. van Dijk, Lilian Linders, Inge van den Akker-Scheek, Adrie Bouma, Leah Schans, Linda Schouten, Patrick Rijnbeek, Rienk Dekker, Martine de Bruijne, Hidde P. van der Ploeg, Willem van Mechelen, Judith G. M. Jelsma.

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
