## [Decision Letter · Decision Letter 0]

4 Feb 2024

PONE-D-23-36328Implementation barriers and facilitators for referral from the hospital to community-based lifestyle interventions from the perspective of lifestyle professionals: a qualitative study.PLOS ONE

Dear Dr. te Loo,

Thank you for submitting your manuscript to PLOS ONE. After careful consideration, we feel that it has merit but does not fully meet PLOS ONE’s publication criteria as it currently stands. Therefore, we invite you to submit a revised version of the manuscript that addresses the points raised during the review process.

We look forward to receiving your revised manuscript.

Kind regards,

Monika Błaszczyszyn

Academic Editor

PLOS ONE

 [This research is part of the LOFIT-project which is funded by The Netherlands Organization for Health Research and Development (ZonMw), grant agreement no. 555003208. Also, the national knowledge and innovation network “Centre Of Expertise for Prevention in Care and Welfare” of the Inholland University of Applied Sciences (Hogeschool Inholland) funded this research.].  

4. One of the noted authors is consortium [LOFIT-consortium]. In addition to naming the author group, please list the individual authors and affiliations within this group in the acknowledgments section of your manuscript. Please also indicate clearly a lead author for this group along with a contact email address.

5. We note that Figure 1 in your submission contain copyrighted images. All PLOS content is published under the Creative Commons Attribution License (CC BY 4.0), which means that the manuscript, images, and Supporting Information files will be freely available online, and any third party is permitted to access, download, copy, distribute, and use these materials in any way, even commercially, with proper attribution. For more information, see our copyright guidelines: http://journals.plos.org/plosone/s/licenses-and-copyright.

Reviewers' comments:

Reviewer's Responses to Questions

**Comments to the Author**

1. Is the manuscript technically sound, and do the data support the conclusions?

Reviewer #1: Yes

Reviewer #2: Yes

2. Has the statistical analysis been performed appropriately and rigorously? 

Reviewer #1: N/A

Reviewer #2: N/A

3. Have the authors made all data underlying the findings in their manuscript fully available?

Reviewer #1: Yes

Reviewer #2: Yes

4. Is the manuscript presented in an intelligible fashion and written in standard English?

Reviewer #1: Yes

Reviewer #2: Yes

5. Review Comments to the Author

Reviewer #1: Comments to author

Overall comment:

The topic is new and of interest to the audience. The barriers and facilitators to the service have been covered comprehensively. In addition, evidence-based strategies have been provided. I generally recommend the paper for publication considering the given comments/suggestions. Also, the paper needs to be revised for some typo errors.

1. The study presents the results of original research.

Yes

2. Results reported have not been published elsewhere.

Yes

3. Experiments, statistics, and other analyses are performed to a high technical standard and are described in sufficient detail.

Need additional information, as suggested.

4. Conclusions are presented in an appropriate fashion and are supported by the data.

No

5. The article is presented in an intelligible fashion and is written in standard English.

Yes, but there are some typos.

6. The research meets all applicable standards for the ethics of experimentation and research integrity.

Yes

7. The article adheres to appropriate reporting guidelines and community standards for data availability.

Yes

Abstract

Under the method section, avoid using abbreviations or provide the full name (CFIR, ERIC) for those unfamiliar with them.

Introduction

Line 87: consider providing some additional information about the I-change model. It would be helpful for the readers to understand what this model is about.

Line 90: you mentioned that lifestyle coaching in the community is less costly. To whom? To the patients or the healthcare system. If it is less costly for the healthcare system, do you need to hire LFO staff? How many? Do you need to implement a certain referral system? Do you need a clinic to meet with the patients? Is it not costly to implement such a service considering all the mentioned resources?

Is there any published cost-effectiveness analysis to support your statement? Please provide further explanation. Also, it would be helpful to know when the service was implemented in the Netherlands. Especially since most of your participants, 61%, had a working experience of less than 5 years.

Line 99: you mentioned that the lifestyle broker has motivational interviewing skills. What is their qualification? E.g., a dietician or other medical specialty? Are there any required experiences?

The procedure of the referral process is not very clear, how and where they discuss the patients needs? Do the patients need to get an appointment by themselves, or do they get referred by their physician to LFO?

Also, it is not clear where and how the patient will receive the CBLI intervention? At their homes, or particular community or primary care clinics? How would the CBLI communicate with the LFO? Do they provide any written progress notes/letters/emails to the LFO?

Please provide additional information for international readers unfamiliar with this service, as their understanding of the nature of the service will help them understand the subsequent sections.

Line 110: who are the stakeholders in this context? The patients, staff of the LFO, and CBLI? Any others? What about the service leaders or implementers? Did you consider interviewing them? If not, do you think this is one of the limitations? You did interview all the stakeholders. Is that a limitation?

Method

Line 114: “we performed semi-structured interviews with various CBLI”. I am confused with using the abbreviation CBLI, which refers to community-based lifestyle interventions, not personnel. I think it is better to use another abbreviation or term, e.g., community-based healthcare providers, and change it throughout the manuscript.

Line 115-117: “Our aim was to explore views, needs and preferences regarding implementation and collaboration with an LFO in these two medical centers”. I prefer to mention your aim once in the introduction. This is a repetition, and you did not mention your participants here.

Line 118-119: Why was the study exempted from the review by the ethical committee? you can add the statement from your supplementary file 5 here, “The research does not fall within the scope of the Medical Research Involving Human Subjects Act (WMO). The reason is that the research does not have a medical-scientific question”. And remove the file S5.

Line 121: the provided Figure 1 is not of high quality, I could not read what is written in it. Please provide a higher-quality figure.

Line 158-159: I do not understand what the BLCN stands for. Also, does GLI stand for Combined Lifestyle Intervention? Or is it CLI? Is it a typo error?

Line 168: what is the necessity of doing a video call interview? Why not a phone call? Isn’t easier?

Line 181: could be? Or some sections were skipped!

Line 186: you introduce the service to the participants; this means that not all of them were familiar with the service. Should you clarify this in your inclusion criteria? E.g., participants were deemed eligible to participate whether they had previous knowledge of the service or not.

Line 202: Were the interviews in English? Or in Dutch, so did you have to translate them? If yes, what was the method of translation of the transcripts? If you did not translate the transcripts, does the used software support the Dutch language?

Line 224: you mentioned that translations were made at the end, does this mean all the previous steps were in Dutch?

Results

Line 241: it would also be helpful to know how many were already familiar with the referral service. They might have more insight into the barriers and facilitators if they witness the early implementation of the service, in which case they can explore barriers/facilitators at the early stage of the implementation and after running the service for several years.

The participant's knowledge of the service was in mind as I read through the quotations.

Line 414: Is the communication between 1ry and 2ry care always via email? Is there any paper referral? I think you need to better explain the service in the introduction, so when the reader comes to this section, he/she will already have enough understanding of the procedure.

You did not mention when you reached the saturation, and what types of saturation was considered in this study? What about triangulation, credibility, and transferability of the data.

Elaborate more on why you choose interview vs. focus group. Did you reach the data saturation? For some professions, you only interviewed 1, like a dietician; how did you reach saturation about the perspectives of this professional? Could consider this in the limitation of the study?

Discussion

Line 582: is the abbreviation correct? Integral Health care Agreement (IZA)?

Line 588: maybe you can discuss the triangulation here “A strength of this study is the input from various lifestyle professionals”

Line 596-597: “The interviews were performed in order to develop the LFO”. It would be better to move this to the introduction. Because the information provided in the introduction gives the reader the impression that the service is already available but partially implemented. This need to be clarified in the introduction and your aim.

Line 598: Did none of the participants have any experience with the service?

Line 599: I appreciate that you used evidence-based tools to develop your strategies and map them with the CFIR framework, but I was expecting to read some examples from previous studies showing that these strategies did work in practice. You did not mention anything about the improvement strategies in the discussion section.

What are the practical implications or recommendations for practice? You may need to write a few lines about it.

Conclusion

The conclusion is not firm and needs some work. You need to discuss the implications for practice and some future work to strengthen your conclusion.

Reviewer #2: Many thanks for the opportunity to review this work. The authors have reported the qualitative study well and made use of implementation science which is good. However, there are a few concerns. It is not clear that the LFO does not exist and that this piece of work is to inform the design and development and implementation of this. The reference to the CFIR is very vague and almost tokenistic. The labelling of the themes are more topic titles rather than descriptions of the themes. The proposition of matching ERIC to the CFIR categorised barriers is not robust. Authors should reconsider what this study has really investigated and demonstrated. This will help to not over-state the findings and implications.

More detailed comments are provided below:

In the abstract results, this is not clear: 'determine qualifications professionals'. Please revise.

In the introduction:'as patients are more motivated for a

82 lifestyle change when experiencing a health event.' Do the authors have any evidence to support this?

The authors make reference to the I-Change model but give no explanation of it and its significance. I would recommend providing a brief outline or removing the mention of this.

The LFO appears to be a novel concept. Is this just being implemented in the researchers' area or is this a national model. It would be good to provide a little bit of information about where this model is being provided and how successful this is if evidence exists.

Please fix the first line in the table where it states 'total = 23242'

It seems the themes might be better labelled:

'LFO knowledge about financial status and/or health eligibility of the patient'

'LFO knowledge about the cultural and linguistic needs of the patient'

'LFO knowledge about the structure and availability of CBLI services'

'LFO knowledge about the quality of CBLI services'

'Agreed referral process and content'

'Secure and equitable communication platform'

'Personal relationships between the LFO and CBLI to facilitate collaboration'

Develop support tools: how should the barrier about keeping up to date with appropriate services available be overcome?

Would a manual about the use of the communication tool be the best option? Would it not be better to find a system-wide digital solution to facilitate easy communication?

Build networks: this is really vague. I recommend the authors reconsider what the network could look like, how this might be established and sustained.

Facilitate learning collaboratives: these strategies appear to present a significant workload for the LFO. Can the authors potentially consider other options that might be more feasible and attractive?

Please can the authors clarify this sentence:' In order to realize this, representatives of the care sport connectors could be included

527 in the development and implementation team before implementation. '

The key concern with the ERIC strategies is that they are proposed by the research team. Not all seem to be very practical or feasible. It is not clear if the research team have sufficient knowledge and expertise to add credibility to recommending these strategies. Because of this, the recommendations have little value.

Discussion: I would argue that the limited number of referral options for low income patients and for non-Dutch speaking people and those living in rural areas and quality of referral options are not barriers for the implementation of a hospital based LFO. These are simply limitations of the current CBLI offer...

What do the authors mean here: '. Matching strategies were developed and for the

larger part implemented. '?

How do the authors know that these were the most important strategies: ' Most important strategies concerned identifying limited referral

options and professionals with a quality hallmark and strategies to streamline the actual

referral and the process of collaboration between basic services, primary and secondary

care'

What is the relevance of this in the context of this study: ' Digital health can facilitate and improve health care in various applications, such as

diagnostic tools, in order to make disease management and communication health care

processes more patient-centered [44, 45].'

The reader becomes aware: 'The interviews were performed in order to develop the LFO.' In the last section of the discussion! This should be made clearer at the beginning, when rationalising the need for the study.

I do not agree with this strength: ' A major strength of this study is that we systematically matched

600 evidence-based implementation strategies to the found barriers of the CBLI.' Given how these strategies were selected, I am unconvinced that these are the most practical, feasible and cost effective recommendations to implement the LFO.

Given this paper is about developing an intervention to be provided at the intersection between primary and secondary care, the discussion should be considering wider literature about other interventions that have been developed and tested at this juncture, e.g. care coordinators, disharge planners. disease-specific interventions if appropriate.

6. PLOS authors have the option to publish the peer review history of their article (what does this mean?). If published, this will include your full peer review and any attached files.

Reviewer #1: No

Reviewer #2: No

---

## [Author Response · Author response to Decision Letter 0]

2 Apr 2024

Thank you for providing solid feedback on how to improve our manuscript. In the document "Response to Reviewers" all points are answered (see below for the content).

Response #1:

We checked the requirements and made minor changes to the file naming of the supplementary files.

 [This research is part of the LOFIT-project which is funded by The Netherlands Organization for Health Research and Development (ZonMw), grant agreement no. 555003208. Also, the national knowledge and innovation network “Centre Of Expertise for Prevention in Care and Welfare” of the Inholland University of Applied Sciences (Hogeschool Inholland) funded this research.]. 

Response #2:

We added the role of the funders in the cover letter.

Response #3:

All relevant data are within the manuscript and its Supporting Information files.

Within Amsterdam UMC it is mandatory to delete audio recordings after transcribing and checking. The (Dutch) transcripts of the interviews, are available upon reasonable request due to ethical and legal restrictions under the General Data Protection Regulation (EU). The study participants did not consent to make the full transcripts of interviews publicly available. 

Excerpts of the raw data (quotes) are available within the text of this paper. Requests for anonymized excerpts from the full transcripts can be send to the general email address: lofit@amsterdamumc.nl

4. One of the noted authors is consortium [LOFIT-consortium]. In addition to naming the author group, please list the individual authors and affiliations within this group in the acknowledgments section of your manuscript. Please also indicate clearly a lead author for this group along with a contact email address.

Response #4: 

in the acknowledgement section the individual authors of the consortium are listed. We added their affiliations and indicated a lead author and email address for the consortium on page 34-35, lines 690-724.

5. We note that Figure 1 in your submission contain copyrighted images. All PLOS content is published under the Creative Commons Attribution License (CC BY 4.0), which means that the manuscript, images, and Supporting Information files will be freely available online, and any third party is permitted to access, download, copy, distribute, and use these materials in any way, even commercially, with proper attribution. For more information, see our copyright guidelines: http://journals.plos.org/plosone/s/licenses-and-copyright.

Response #5: we have altered the figure so that it complies with the CC BY 4.0 license. The current images that are in the figure have been made especially for our study.

Reviewer #1: Comments to author

Abstract

Under the method section, avoid using abbreviations or provide the full name (CFIR, ERIC) for those unfamiliar with them.

Response #6:

Thanks for this solid advice. We have provided the full names of the abbreviations (the consolidated framework for implementation research (CFIR) in combination with the Expert Recommendations for Implementing Change (ERIC) strategies) in the revised manuscript with track changes on page 3 , line 51-53.

Introduction

Line 87: consider providing some additional information about the I-change model. It would be helpful for the readers to understand what this model is about.

Response #7:

We named the I-Change model to support the idea that different interacting factors lead to health behavior. This model is one of many and because we did not use it further in the study, we can see that it is confusing to just name this one. Because of this we removed the name of the model in the text on page 5, line 92; which is in agreement with the advice of reviewer 2 as well.

Line 90: you mentioned that lifestyle coaching in the community is less costly. To whom? To the patients or the healthcare system. If it is less costly for the healthcare system, do you need to hire LFO staff? How many? Do you need to implement a certain referral system? Do you need a clinic to meet with the patients? Is it not costly to implement such a service considering all the mentioned resources?

Is there any published cost-effectiveness analysis to support your statement? Please provide further explanation. Also, it would be helpful to know when the service was implemented in the Netherlands. Especially since most of your participants, 61%, had a working experience of less than 5 years.

Response #8: 

What was meant here is that lifestyle coaching in the community is less costly for the healthcare system than lifestyle care in the hospital, due to the complexity of care in an UMC. 

In the Netherlands health care (in the hospital and in the community) is financially almost completely covered by basic health insurance. Citizens are required to have basic health care insurance for which they pay a monthly contribution. When patients use care, they additionally pay a maximized yearly excess. The Dutch government aims to keep this system affordable by designating care at the best quality-cost ratio, which means that when possible, care is given in the community instead of the hospital. Ultimately this flows back to the patients as well, as expensive care will lead to higher basic health insurance contribution.

The current study served to develop the LFO, which was not yet existing at that time. Using the results of this study, we implemented the LFO and simultaneously started a (cost) effectiveness study. We agree with the reviewers that the manuscript does not clearly state that the LFO is a novel care concept that still had to be developed. To clarify this, we have made changes to the manuscript on page 3, line 37 ; page 6, line 102-104; page 7, line 121; page 8, lines 127-128 to make this clearer.

The reviewer rightly states that it can be costly to implement an LFO in a hospital, given the resources required. We don't yet know how expensive it is and whether it outweighs the benefits. At this moment, we are executing an RCT to establish the effectiveness and cost-effectiveness of the LFO. The RCT is still ongoing, therefore we have not yet published results.

Line 99: you mentioned that the lifestyle broker has motivational interviewing skills. What is their qualification? E.g., a dietician or other medical specialty? Are there any required experiences?

The procedure of the referral process is not very clear, how and where they discuss the patients needs? Do the patients need to get an appointment by themselves, or do they get referred by their physician to LFO?

Also, it is not clear where and how the patient will receive the CBLI intervention? At their homes, or particular community or primary care clinics? How would the CBLI communicate with the LFO? Do they provide any written progress notes/letters/emails to the LFO?

Please provide additional information for international readers unfamiliar with this service, as their understanding of the nature of the service will help them understand the subsequent sections.

Response #9: Thank you for pointing out that this section needs clarification. We adjusted this section with more information on the procedure (page 6, lines 105-110). However, finding out how the exact procedure of the referral process should be, was part of this research; since we wanted to align as much as possible to the working processes and the preferences of the community-based lifestyle interventions (CBLI). We added this information on page 6, lines 118-120.

Line 110: who are the stakeholders in this context? The patients, staff of the LFO, and CBLI? Any others? What about the service leaders or implementers? Did you consider interviewing them? If not, do you think this is one of the limitations? You did interview all the stakeholders. Is that a limitation?

Response #10: We agree with the reviewer that for successful implementation we must include the perspectives of all stakeholders. In this study, we focused on the external stakeholders of the LFO: the CBLI. Our aim was to design our process and procedures in alignment with the views of these stakeholders. In another study, we focused on the internal stakeholders of the LFO: the health care professionals in the hospital and also the patients. We added this information in the discussion on page 32, line 664- 667. The research team was and is also responsible for the development and implementation of the LFO. In this way, we have taken into account the perspectives of all relevant stakeholders.

Method

Line 114: “we performed semi-structured interviews with various CBLI”. I am confused with using the abbreviation CBLI, which refers to community-based lifestyle interventions, not personnel. I think it is better to use another abbreviation or term, e.g., community-based healthcare providers, and change it throughout the manuscript.

Response #11: we agree, that is confusing. The CBLI are lifestyle interventions that the LFO can refer to. We interviewed professionals that work for organizations that provide these lifestyle interventions. To make it not more confusing by adding another new term and abbreviation, we decided to add the word ‘professional’ after CBLI wherever we are referring to the person and not to the intervention.

Line 115-117: “Our aim was to explore views, needs and preferences regarding implementation and collaboration with an LFO in these two medical centers”. I prefer to mention your aim once in the introduction. This is a repetition, and you did not mention your participants here.

Response #12: we agree. We removed the aim here.

Line 118-119: Why was the study exempted from the review by the ethical committee? you can add the statement from your supplementary file 5 here, “The research does not fall within the scope of the Medical Research Involving Human Subjects Act (WMO). The reason is that the research does not have a medical-scientific question”. And remove the file S5.

Response #13: thanks for the suggestion, we did as such. 

Line 121: the provided Figure 1 is not of high quality, I could not read what is written in it. Please provide a higher-quality figure.

Response #14: we changed the figure and provided a higher-quality figure

Line 158-159: I do not understand what the BLCN stands for. Also, does GLI stand for Combined Lifestyle Intervention? Or is it CLI? Is it a typo error?

Response #15: these were the Dutch abbreviations. We replaced them by English abbreviations: (Association for Lifestyle Coaches in the Netherlands (ALCN); Combined Lifestyle Intervention (CLI)) (page 10, line 178-180).

Line 168: what is the necessity of doing a video call interview? Why not a phone call? Isn’t easier?

Response #16: these interviews were held during the time that COVID-19 was limiting the possibility for physical meetings, which we would have preferred for the interviews in order to be able to fully engage with the respondent. A video call was in our view the best alternative to physical meetings. While telephone calls would have been possible and perhaps easier, we believed they would not have provided the same connection with, and in-depth information from, the respondents. 

Line 181: could be? Or some sections were skipped!

Response #17: the reviewer is correct; some sections were skipped depending on the job title. We changed that (page 11, line 202).

Line 186: you introduce the service to the participants; this means that not all of them were familiar with the service. Should you clarify this in your inclusion criteria? E.g., participants were deemed eligible to participate whether they had previous knowledge of the service or not.

Response #18: we recognize that we did not state clearly enough in the introduction that the service (LFO) did not yet exist at all. We hope that it is now much clearer due to the changes we made (see response #8, third paragraph).

Line 202: Were the interviews in Engli

---

## [Decision Letter · Decision Letter 1]

7 May 2024

Implementation barriers and facilitators for referral from the hospital to community-based lifestyle interventions from the perspective of lifestyle professionals: a qualitative study.

PONE-D-23-36328R1

Dear Dr. Leonie Mariëlle te Loo,

We’re pleased to inform you that your manuscript has been judged scientifically suitable for publication and will be formally accepted for publication once it meets all outstanding technical requirements.

Kind regards,

Monika Błaszczyszyn

Academic Editor

PLOS ONE

Additional Editor Comments (optional):

Reviewers' comments:

Reviewer's Responses to Questions

**Comments to the Author**

1. If the authors have adequately addressed your comments raised in a previous round of review and you feel that this manuscript is now acceptable for publication, you may indicate that here to bypass the “Comments to the Author” section, enter your conflict of interest statement in the “Confidential to Editor” section, and submit your "Accept" recommendation.

Reviewer #1: All comments have been addressed

2. Is the manuscript technically sound, and do the data support the conclusions?

Reviewer #1: Yes

3. Has the statistical analysis been performed appropriately and rigorously? 

Reviewer #1: Yes

4. Have the authors made all data underlying the findings in their manuscript fully available?

Reviewer #1: Yes

5. Is the manuscript presented in an intelligible fashion and written in standard English?

Reviewer #1: Yes

6. Review Comments to the Author

Reviewer #1: Thanks for addressing all the comments and enhancing the paper. It is a good paper with a new, exciting concept.

My only comment is that I think the manuscript needs to be revised regarding punctuation (some full stops were missing in some paragraphs). No comments on the scientific information.

Best of luck

7. PLOS authors have the option to publish the peer review history of their article (what does this mean?). If published, this will include your full peer review and any attached files.

Reviewer #1: No

---

## [Editor Report · Acceptance letter]

13 May 2024

PONE-D-23-36328R1 

PLOS ONE

Dear Dr. te Loo, 

I'm pleased to inform you that your manuscript has been deemed suitable for publication in PLOS ONE. Congratulations! Your manuscript is now being handed over to our production team.

Kind regards, 

on behalf of

Dr. Monika Błaszczyszyn 

Academic Editor

PLOS ONE